# High Reynolds number wind turbine blade equipped with root spoilers. Part I: Unsteady aerodynamic analysis using URANS simulations.

Thomas Potentier[1,3], Emmanuel Guilmineau[1], Arthur Finez[2], Colin Le Bourdat[3], and Caroline Braud[1]

[1]LHEEA, 1 rue de la Noë 44321 Nantes Cedex 3 - FRANCE
[2]ENGIE Green, 59 Rue Denuzière 69002 Lyon - FRANCE
[3]ENGIE Green, 15 rue Nina Simone 44000 Nantes - FRANCE

**Correspondence:** Thomas Potentier (thomas.potentier@ec-nantes.fr)

**Abstract.** A commercial wind turbine blade equipped with root spoilers is analysed using 2D URANS Computational Fluid Dynamics (CFD) to assess the unsteady impact of passive devices. In this work, we present the 2D CFD unsteady results from a non-rotating single thick section located at the root end of the blade with and without spoiler. Computations were performed at the chord-based Reynolds number $Re_c = 3 \times 10^6$. The analysed spoiler is of commercial size with a height of approximately 33% of the local chord. Comparatively to the existing literature, it is at least one order of magnitude larger than the size of the well known Gurney flaps. The analysis is first performed in the steady state at a single angle of attack using global aerodynamic forces, the local pressure distributions and flow field analysis. It shows a very important flow rearrangement in the presence of a spoiler, responsible of the mean lift force enhancement. Then, simulations are extended over a large range of angle of attack (from $-20°$ to $20°$), to identify the spoiler behaviour across the polar. Analyses are then continued accounting for the flow unsteadiness. The spoiler induces an important wake behaviour linked to the apparition of global load fluctuations. Using the wall pressure distributions and the associated spatio-temporal organisation of the flow field those fluctuations are well characterised. The detailed analysis performed at one angle of attack, is then extended to assesses the load fluctuations at other angles of attack, showing the evolution of the unsteady loads in relation to the wake.

## 1 Introduction

Wind energy, over the last decades, increased its market share thanks to longer blades and a continuous increase in rated power. Nevertheless, to keep lowering the Levelised Cost of Energy (LCoE), onshore turbines need to produce more energy within the same swept area. Indeed, the blade size is restricted to avoid (or limit), among other things: acoustic emission, aeronautical interference, local population rejection... The blade design imposes high blade thickness at the root end of the blade for structural reasons, it leads to significant loss of the aerodynamic performances. It is detrimental to the energy extraction, therefore solutions were developed to improve this blade region, among them, the passive Aerodynamic Add-Ons (AAO).

AAO are devices attached to the blade surface to either increase the power extraction locally or reduce the acoustic emission of the turbine and thereby allowing the exploiting party to use the full turbine's capacity. The current paper will focus on the

former type of AAO: passive AAO installed in the root blade area to improve the aerodynamic performances of the thick aerofoil profile types. The AAO solution has been explored by many authors before (see Pechlivanoglou (2013); Saleem (2019); Bach (2016)). It is to be noted that in opposition to passive devices, some active solutions exist in the research state (such as solutions described in: Jaunet and Braud (2018); Boeije et al. (2009); McWilliam et al. (2018)) but are not yet available to the market.

The flow behind two well-known AAO devices, the Vortex Generators (VG) and the Gurney Flaps (GF), has been largely investigated in order to understand its mechanism and control benefit which is summarised hereafter. VG are small fins (thin plates of usually triangular or rectangular shapes attached to a base plate) attached on the aerofoil suction side to delay stall by re-energising the boundary layer (see Taylor et al. (1947); Godard and Stanislas (2006); Cathalifaud et al. (2009); De Tavernier et al. (2021); Gao et al. (2014); Lin (2002); Skrzypiński et al. (2014); Perivolaris and Voutsinas (2001)). The vortices, aligned with the inflow leaving the device, increase the mixing between high speed flow (free stream) and low speed flow (boundary layer) thus delaying the flow separation (see Schubauer and Spangenberg (1960)). GF are devices installed at the aerofoil trailing edge on the pressure side. They aim to create an artificial camber seen by the flow, it will decrease the lower pressure on the suction side and therefore increase the pressure difference between both sides of the aerofoil thus increasing the generated lift (see Liebeck (1978); Cole et al. (2013); Giguere et al. (1995); Wang et al. (2008); Jang et al. (1998); Sørensen et al. (2014); Mohammadi et al. (2012); Li et al. (2002); Alber et al. (2020); Meena et al. (2017)). The main difficulty in AAO design is the correct sizing: for GF too small and the gain is nonexistent, too large and the drag penalty cannot be compensated by the lift gain. Similarly for VGs, geometrical parameters of the device are affecting the control efficiency. The combined effect of both VG and GF solutions seems to be additive according to Storms and Jang (1994).

Megawatt size wind turbines are experiencing high Reynolds number ($Re > 10^6$) for high relative aerofoil thicknesses at the blade root (relative thickness $> 36\%$), whereas most of the literature available is either targeting thin aerofoils at generally low to moderate Reynolds number, or small AAO sizes generally within the boundary layer thickness. Interesting outputs from Meena et al. (2017) could be drawn such as a detailed characterisation of the shedding vortex types occurring behind aerofoils equipped with different GF heights using Unsteady Reynolds-Averaged Navier-Stokes (URANS) and Large Eddy Simulations. However, the simulations were limited to thin profiles and low Reynolds numbers. The present study will contribute to extend such work towards thick profiles and higher Reynolds numbers. Also, there exist a gap on the AAO size (usually $\geq 5\%$ of blade chord) used by manufacturers and the one available in the literature (up to 1% of blade chord). Another objective of the present paper is to contribute to reduce this gap. The AAO solution presented in this paper is the spoiler, it is a passive obstacle installed on the aerofoil pressure side to increase the aerofoil camber perceived by the flow. Despite a lift enhancing mechanism similar to the large GF one, there is a main difference which lies in its position. The spoiler is installed between 60% and 80% of the local chord and is a long single aerodynamically designed part while the GF is installed perpendicularly to the local chord at the trailing edge and is usually comprised of several smaller parts butted up together.

The European AVATAR project (see Schepers (2017)) studied thick sections at realistic Reynolds number ($Re > 10^6$). According to the authors' knowledge, it is the only literature about wind turbine blade root spoilers at realistic operating conditions. 2D Computational Fluid Dynamics (CFD) simulations were performed on the blade root with and without spoiler using Reynolds

Average Navier-Stokes (RANS) thereby smoothing out the possible unsteady effects.

The present paper is therefore dedicated to the analysis of high Reynolds unsteady simulation, $Re_c = 3 \times 10^6$, using a scanned blade shape from an operating 2MW turbine. Two configurations will be analysed: bare blade (no AAO) and a large spoiler existing on an ENGIE Green turbine. The methodology to post-process the scanned blade with the spoiler is explained in Section 2, the CFD set-up and mesh independence study is described in Section 3, finally, analysis of the AAO impacts in term of steady and unsteady aerodynamics properties are presented in detailed in the result Section 4.

## 2    Methodology

### 2.1    Blade root section with and without spoiler

The wind turbine geometry used in the present study was acquired during a scanning campaign on an operating 2MW turbine (see Dambrine (2010)). Scanned cross sections were created by extracting thin slices of 5cm wide of the entire point cloud. Then, we post-treated each cross-section by ordering the point cloud coordinates and fitted splines. Several interpolation tech-
niques were tried, the Bezier curves gave the best outcome, resulting in a smooth and continuous geometry despite missing scan data due to the scanner position not being able to capture some areas of the blade (see Figure 1). Nevertheless, small discrepancies (peaks) are present at $x/c \approx 10\%$ and $x/c \approx 60\%$ due to residual panelisation with however negligible on the lift evaluation.

The scanned blade was originally equipped with root spoilers. The blade without spoiler was generated by manually removing
parts of the cloud points corresponding to the spoiler location (see Figure 2), consequently wherever the spoiler is not present both geometry are identical. The unsteady analysis will focus on a single radius at the radial position 6m from the blade root: $\frac{r}{R} = 13\%$, $r$ is the local radius and $R$ is the blade length, located in the middle of the spoiler. This location was chosen to minimise 3D effects from the spoiler ends, thus ensuring a closer representation of reality when simulated in 2D.

## 3    CFD computations

### 3.1    CFD solver

ISIS-CFD, developed by Centrale Nantes and CNRS and available as a part of the FINE™/Marine computing suite, is used in the present study to solve the incompressible Unsteady Reynolds-Averaged Navier-Stokes (URANS) equations. It is based on the finite volume method to build the spatial discretization of the transport equations. The unstructured discretization is face-based, which means that cells with an arbitrary number of arbitrarily shaped faces are accepted (unstructured mesh). A
second order backward difference scheme is used to discretize time. The solver can simulate both steady and unsteady flows. In the case of turbulent flows, transport equations for the variables in the turbulence model are added to the discretization. All flow variables are stored at the geometric cent of arbitrary shaped cells. Volume and surface integrals are evaluated with

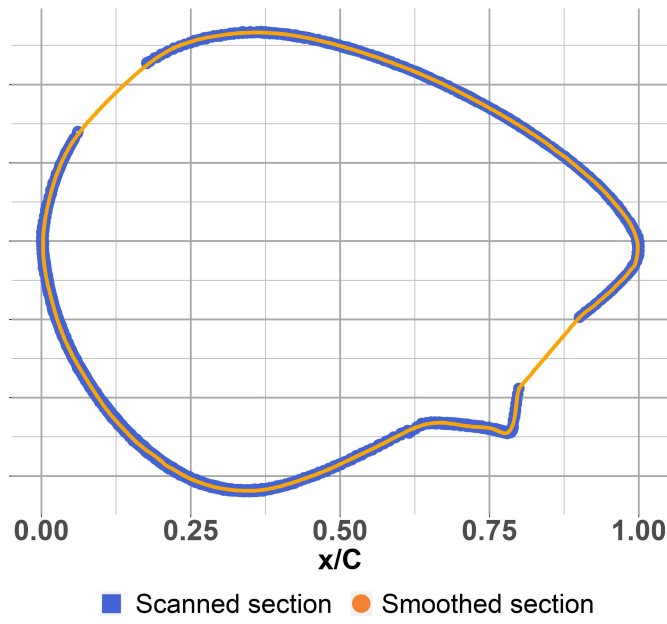

**Figure 1.** Scanned section and smoothed section at radial position R6 ($\frac{r}{R} = 13\%$). The blue square (■) shows the scanned point cloud while the orange dot (●) shows the smoothed section.

second-order accurate approximations. The method is face-based, which means that the net fluxes in the cells are computed face by face. Thus, the cells with an arbitrary number of arbitrarily shaped faces are accepted. Numerical fluxes are reconstructed
on the mesh faces by linear extrapolation of the integrand from the neighbouring cell centres. A centred scheme is used for the diffusion terms, whereas for the convective fluxes, a blended scheme with 80% central and 20% upwind is used.

The velocity field is obtained from the momentum conservation equations and the pressure field is extracted from the mass equation constraint, or continuity equation, transformed into a pressure equation. The pressure equation is obtained by the Rhie and Chow interpolation Rhie and Chow (1983). The momentum and pressure equations are solved in an segregated manner as
in the SIMPLE coupling procedure Issa (1986). A detailed description of the discretisation is given by Queutey and Visonneau (2007).

The turbulence model used is SST k-$\omega$ (see Menter (1993)). The flow characteristics are representing the air at sea level at a temperature of 15°C, i.e.: $\nu = 1.81 \times 10^{-5}$kg m$^{-1}$ s$^{-1}$ (dynamic viscosity) and $\rho = 1.225$kg m$^{-3}$ (air density). A uniform inflow of 45m/s is set, which induces a chord Reynolds number of: $Re_c = 3 \times 10^6$ for the presented section of $\frac{r}{R} = 13\%$.

## 3.2 Boundary conditions and mesh independence

A comparison between the existing DANAERO literature both experimental and numerical and ISIS-CFD was performed (see Potentier et al. (2020)). The outcome showed the good agreement between the 2D wind tunnel experiment and the 2D URANS

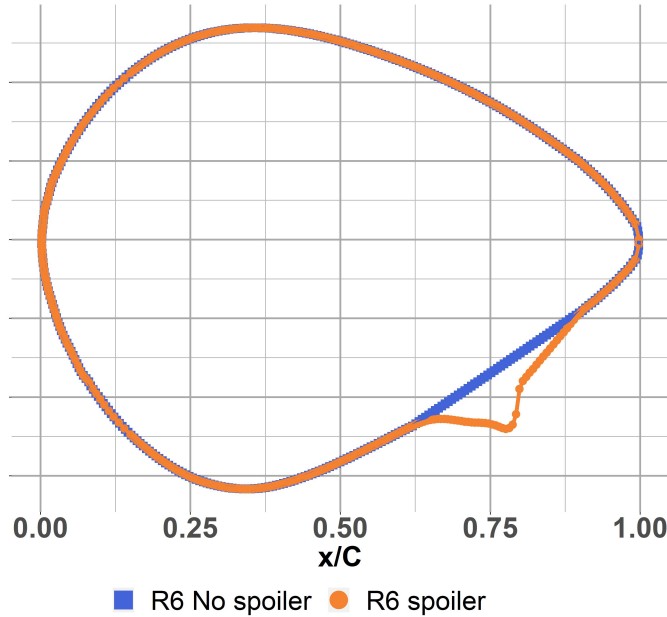

**Figure 2.** Overimposed aerofoil shapes at radial position R6 ($\frac{r}{R} = 13\%$): the blue square (■) shows the no spoiler coordinates while the orange dot (●) shows the spoiler coordinates

ISIS CFD simulations, thus validating the use of ISIS-CFD for 2D external applications. The domain size study has also been performed and the recommended square domain of 80 chords in length was used. The aerofoil related surfaces boundary conditions were described as "no slip wall". The free stream velocity condition was imposed on the inlet, upper and lower boundaries and the outlet boundary was using the "prescribed pressure" condition. Finally, $y^+ = 0.15$ was imposed on the aerofoil surfaces and the automatic grid refinement feature was activated so as to track more accurately the wake vortices (see Wackers et al. (2014, 2017)). The "no spoiler" aerofoil is originally described with 362 pairs of [X;Y] coordinates and the "spoiler" with 503. The leading edges are both positioned at [0;0]. A convergence study was carried out using the lift and drag coefficients, during the mesh refinement the number of faces defining the aerofoil geometry changed as described in Table 1 and Table 2. FINE™/Marine provides the time series for the lift ($L$) and drag ($D$) evolution, the respective coefficients are calculated by the equation 1.

$$C_L = \frac{2 \times L}{\rho c U_\infty^2} \qquad\qquad C_D = \frac{2 \times D}{\rho c U_\infty^2} \qquad\qquad (1)$$

Where $U_\infty$ is the relative velocity of 45m/s and $c$ the aerofoil chord. Four meshes were tested to assess the grid independence: Coarse, Medium, Fine and Very fine. Both cases used the same input conditions for the viscous layers insertion and automatic grid refinement for each mesh. Because the calculations were performed using the automatic grid refinement, the Richardson extrapolation is calculated using the final mesh configuration.

The results in Table 1 and Table 2 show that the grid is independent both in $C_L$ and $C_D$. The error between the "Very fine" and "Fine" mesh is small enough to be acceptable. For the rest of the study, the "Fine" mesh will be used.

**Table 1.** Grid independence study for the scanned blade without spoiler at $\alpha = 0°$ and $Re_c = 3 \times 10^6$.

| Mesh type | Domain cell count | Aerofoil faces count | $C_L$ | $C_L$ error | $C_D$ | $C_D$ error |
|---|---|---|---|---|---|---|
| Coarse | 44 298 | 459 | 0.319 | -10.28% | 0.07697 | 4.05% |
| Medium | 71 205 | 675 | 0.342 | -3.97% | 0.07622 | 3.04% |
| Fine | 104 907 | 1048 | 0.346 | -2.70% | 0.07488 | 1.23% |
| Very fine | 176 921 | 1535 | 0.355 | -0.41% | 0.07486 | 1.19% |
| Richardson extrapolation | $\infty$ | NA | 0.356 | | 0.07397 | |

**Table 2.** Grid independence study for the scanned blade with spoiler at $\alpha = 0°$ and $Re_c = 3 \times 10^6$.

| Mesh type | Domain cell count | Aerofoil faces count | $C_L$ | $C_L$ error | $C_D$ | $C_D$ error |
|---|---|---|---|---|---|---|
| Coarse | 54 543 | 527 | 0.658 | 6.04% | 0.09015 | 5.62% |
| Medium | 82 543 | 733 | 0.630 | 1.63% | 0.08740 | 2.39% |
| Fine | 137 122 | 1085 | 0.619 | -0.18% | 0.08705 | 1.98% |
| Very fine | 227 686 | 1591 | 0.620 | -0.03% | 0.08584 | 0.57% |
| Richardson extrapolation | $\infty$ | NA | 0.620 | | 0.08536 | |

A time step convergence study using the "Fine grid" has also been performed and summarised in Table 3. The chosen time step for the rest of the study is $\Delta t = 4.44 \times 10^{-5} s$ because of the good balance between result accuracy and rapidity to achieve convergence.

**Table 3.** Time step independence study for the scanned blade with spoiler at $\alpha = 0°$ and $Re_c = 3 \times 10^6$.

| Time step [s] | $C_L$ | $C_L$ error | $C_D$ | $C_D$ error | Time before convergence [min] |
|---|---|---|---|---|---|
| $2.22 \times 10^{-4}$ | 0.596 | -5.70% | 0.08280 | -6.18% | 2719 |
| $8.89 \times 10^{-5}$ | 0.599 | -5.17% | 0.08289 | -6.08% | 3028 |
| $4.44 \times 10^{-5}$ | 0.619 | -1.99% | 0.08705 | -1.36% | 3709 |
| $2.22 \times 10^{-5}$ | 0.628 | -0.54% | 0.08809 | -0.17% | 11118 |
| Richardson extrapolation | 0.632 | 0.00% | 0.08825 | 0.00% | NA |

# 4 Results

The impact of the spoiler previously described (see Section 2.1) is analysed in this section at the blade location, $\frac{r}{R} = 13\%$. It
will be done using URANS simulations from the ISIS-CFD solver described in Section 3. The steady and unsteady simulation
outcomes with and without spoiler are compared in term of aerodynamic forces, local pressure, velocity distribution, Power
Spectral Density and instantaneous spatial vortex organisation in Section 4.1 and 4.2.

## 4.1 Steady aerodynamics

A comprehension of averaged phenomena is necessary before analysing the unsteady behaviour. We will first focus on the
angle of attack $\alpha = 6°$, in the linear part of the lift curve, where aerofoils usually operate on a megawatt size turbine. Also, it
illustrates a the first noticeable unsteadiness in the flow, which will be detailed later.

### 4.1.1 Mean flow reorganisation

The high velocity region on the upper side (Figure 3), associated to a low field pressure level (Figure 4) exhibits a longer
overspeed area over the upper side for the "spoiler" case than for the "no spoiler" case. It induces a longer and stronger
negative pressure (see Figure 4), in good qualitative agreement with the steady results from Gonzalez-Salcedo (2016). On the
contrary, on the lower side, the high velocity region (Figure 3) is more important for the "no spoiler" case than for the "spoiler"
case, inducing a larger negative pressure region on the lower side of the aerofoil.

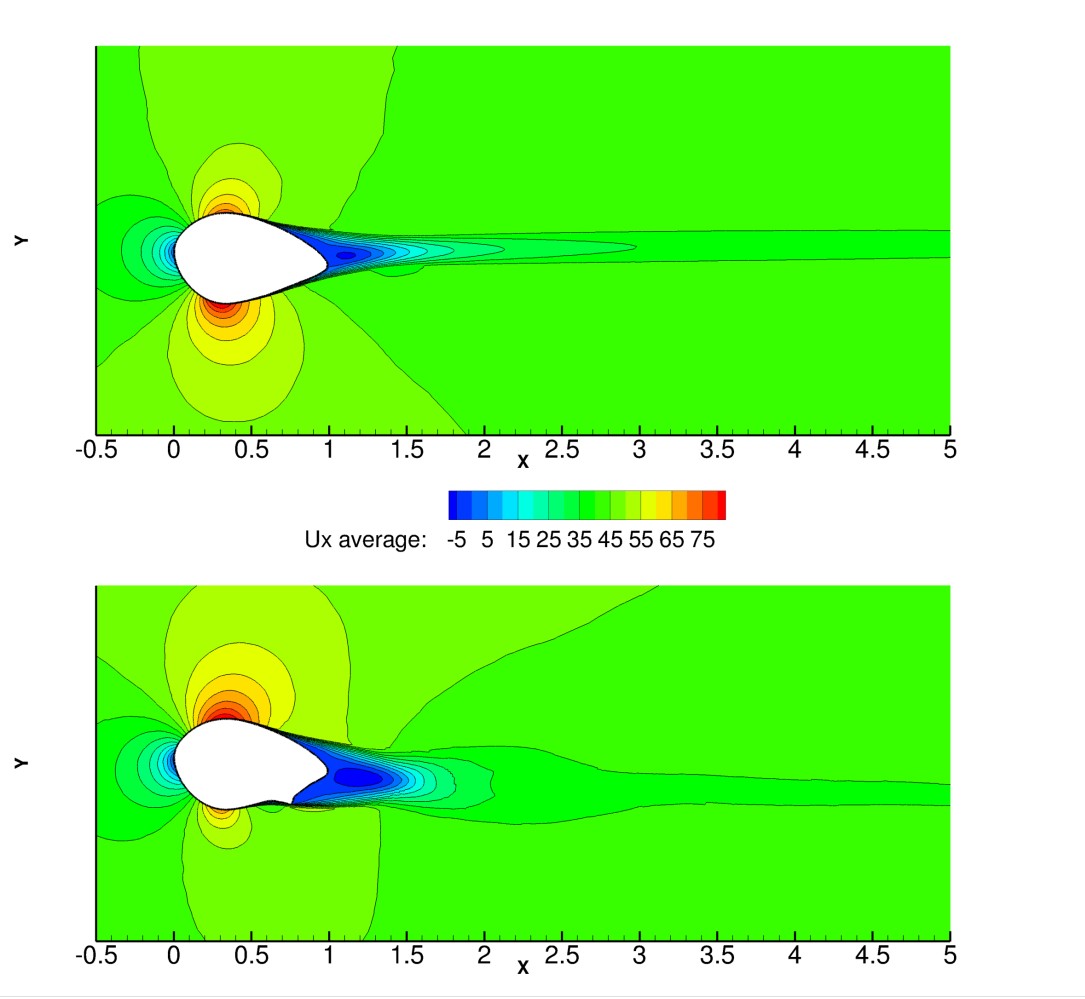

**Figure 3.** Average horizontal wind speed contour plot for $\alpha = 6°$ and $Re_c = 3 \times 10^6$: top - no spoiler case, bottom - spoiler case.

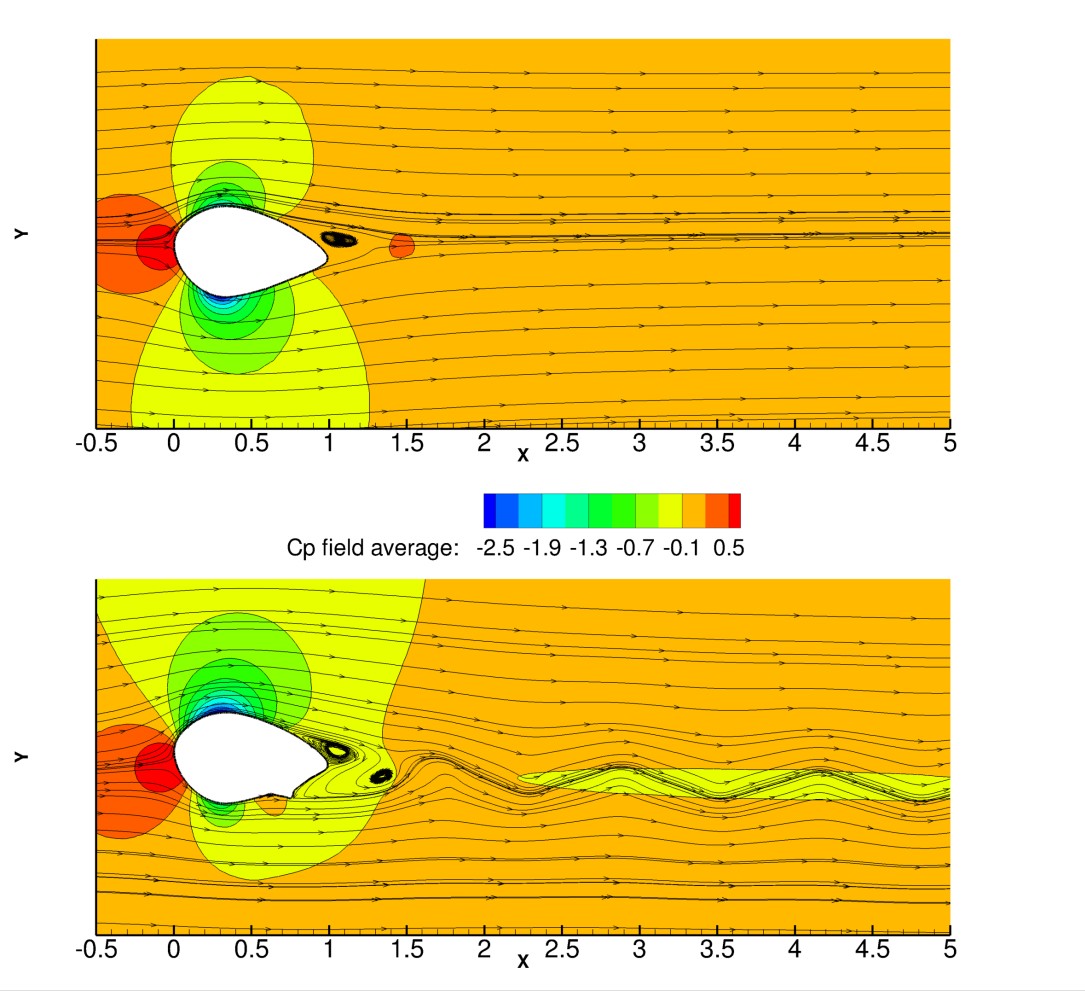

**Figure 4.** Average pressure field contour plot and instantaneous velocity streamlines for $\alpha = 6°$ and $Re_c = 3 \times 10^6$: top - no spoiler case, bottom - spoiler case.

At the wall, the associated pressure coefficient ($C_p$) clearly show that the aerofoil with spoiler has a distribution closer to
thinner aerofoils, with a much larger net area area between the upper and lower curves, and thus a much larger lift than the
reference case (see Figure 5b).

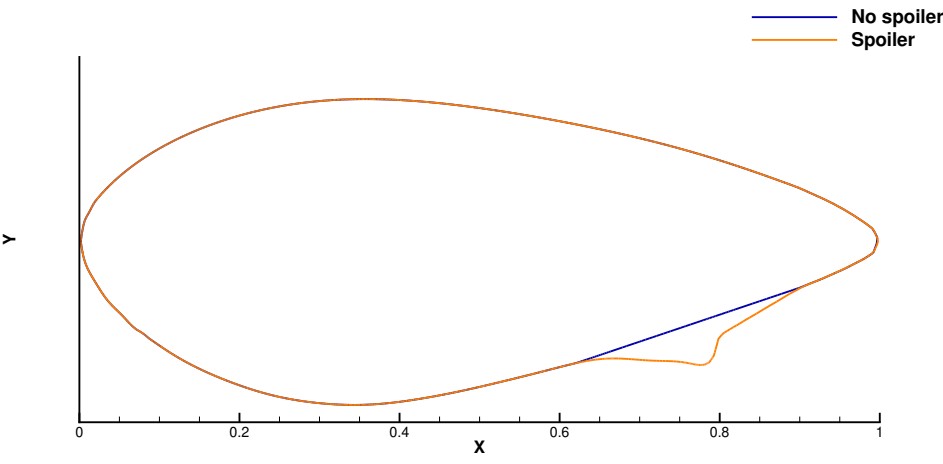

(a) Comparison of the aerofoil shape with and without spoiler.

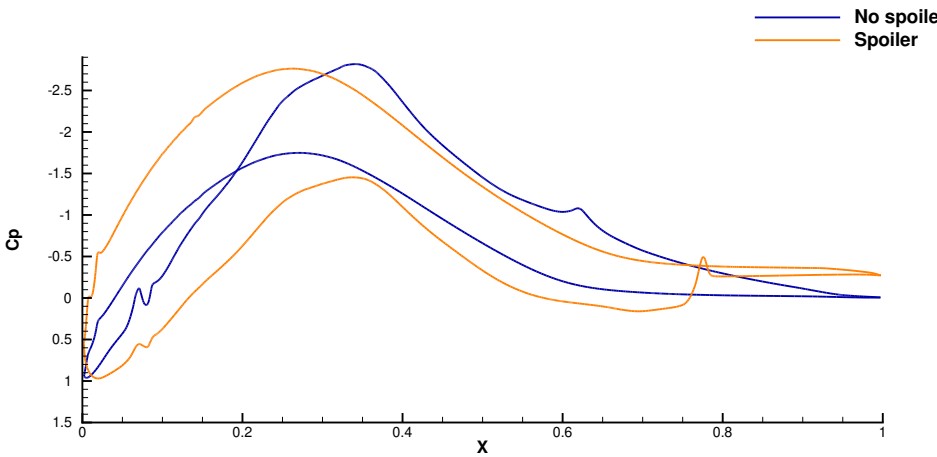

(b) Average wall pressure coefficient plot for $\alpha = 6°$ and $Re_c = 3 \times 10^6$.

**Figure 5.** Illustration of the aerofoils shapes along with the mean wall pressure coefficient.

### 4.1.2 Steady aerodynamic polar

For $\alpha = 6°$, the lift gain when adding a spoiler is $\Delta C_L = 1.34$. This gain however varies with the angle of attack, as can be seen in Figure 6. For the "no spoiler" case, between -4° and 10°, the $C_L$ is decreasing in the usually called "linear region" to reach negative values. This phenomena has been reported by Schaffarczyk and Arakawa (2020) where they analysed a symmetrical thick profile without spoiler at a higher Reynolds number ($Re > 6 \times 10^6$) but the behaviour was similar. Between 10° and 14° the $C_L$ is increasing along with the $C_D$. Beyond 14° both aerodynamic coefficients exhibit a bluff-body behaviour. Whereas, for the "spoiler" case, the lift behaviour is more usual for such Reynolds number: a clear negative stall in the vicinity of -4° can be seen and a positive stall around 8°, despite the constant $C_L$ increase.

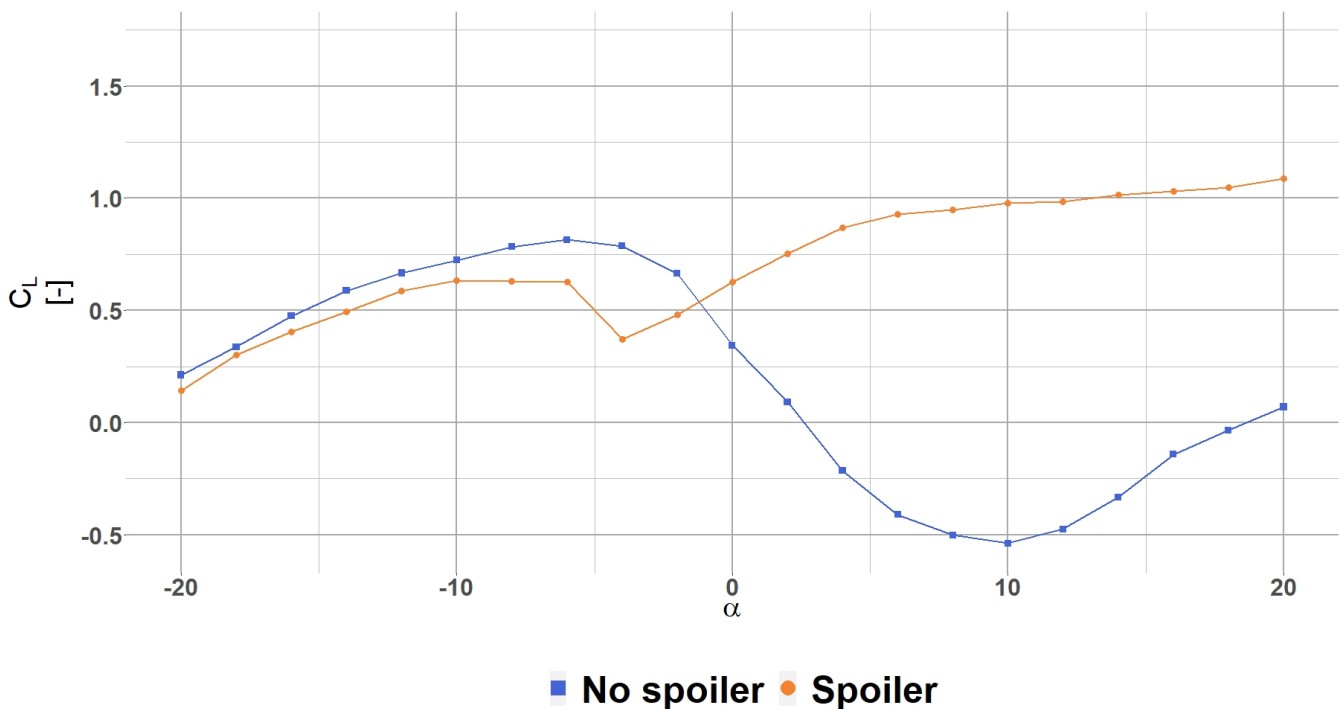

**Figure 6.** Lift coefficient polar for the radial position R6 ($\frac{r}{R} = 13\%$). The blue square (■) shows the $C_L$ for the no spoiler case and the orange dot (●) shows the $C_L$ for the spoiler case.

In the wake region, the mean streamwise velocity component, $U_x$, shows that the mean recirculating area (negative stream-wise velocity) behind the aerofoil with spoiler is wider and extends further downstream (see Figure 3) compared to the "no spoiler" case. This larger wake reflects a drag penalty generated by the spoiler addition that is found of the order of $\Delta C_D = 0.0825$ for $\alpha = 6°$. Again, the penalty is highly dependent on the angle of attack (see Figure 7). There is almost no drag penalty at low angle of attack, up to $\alpha = 0°$. Beyond, the "spoiler" operates at a significantly higher $C_D$ than the "no

spoiler" case.

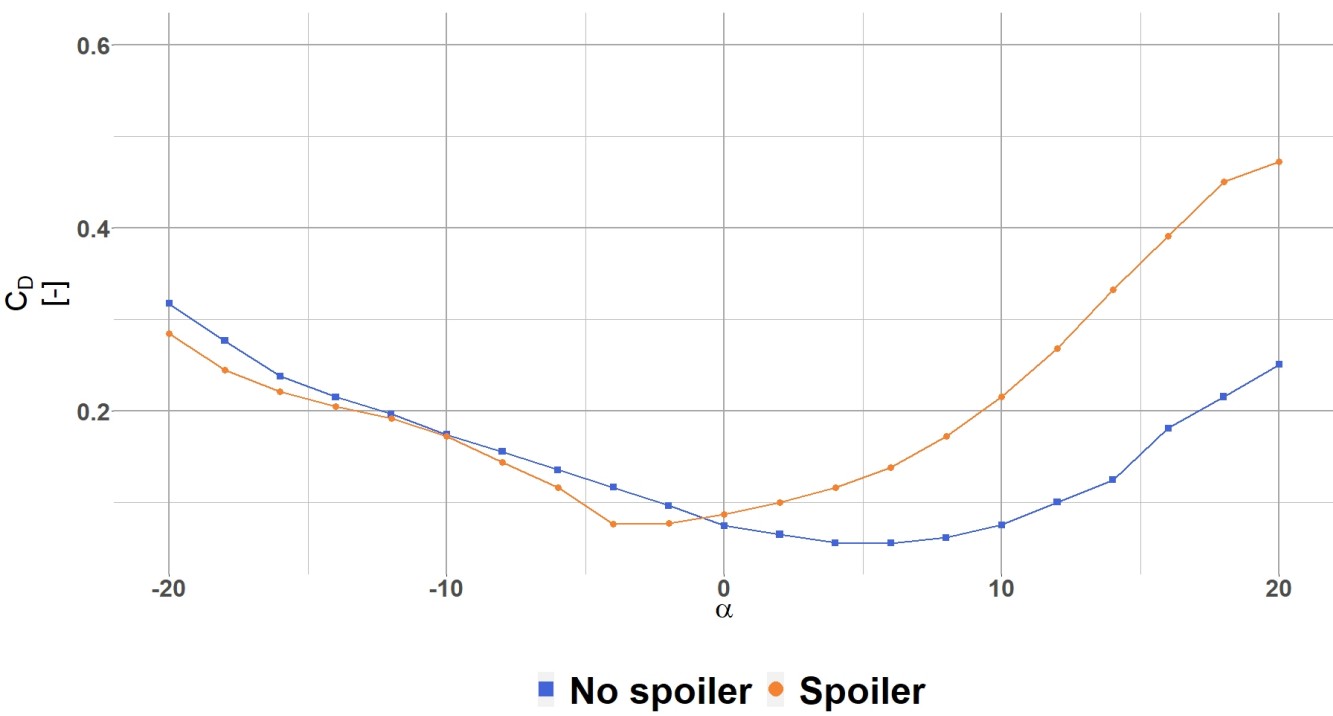

**Figure 7.** Drag coefficient polar for the radial position R6 ($\frac{r}{R} = 13\%$). The blue square (■) shows the $C_D$ for the no spoiler case and the orange dot (●) shows the $C_D$ for the spoiler case.

     In summary, the impact of the spoiler is to redistribute global forces so that the thick root sections become more efficient in term of lift force at a cost of a drag penalty. This known conclusion is in good agreement with the literature of sub-boundary layer GF, except that the lift gain and the drag penalty are much more important and quantified here ($\Delta C_L = 1.34$ and $\Delta C_D = 0.0825$ for $\alpha = 6°$). Globally, in terms of lift gain, adding a spoiler is found detrimental for the negative angles

of attack while of high interest for higher angles of attack. Another drawback of the spoiler addition are the unsteady effects such as shown by the waviness in the instantaneous streamlines behind the "spoiler" case (see Figure 4) and will be detailed in the following section. To the authors' knowledge, the unsteadiness behind large devices at high Reynolds number has not been evaluated, Unsteady Reynolds Averaged Navier-Stokes results of such phenomena are detailed in the next Section 4.2.

## 4.2 Unsteady aerodynamics

The unsteady flow organisation behind thick aerofoil profiles with or without AAOs at high Reynolds number is studied here. The flow analysis focuses on such configuration for one angle of attack first ($\alpha = 6°$), then the aerodynamic polar ranging from $-20° < \alpha < 20°$ will be presented. Unsteady impacts of AAO mostly focuses on the wake region, which will be analysed further in term of instantaneous vorticity, Q-criteria, local (wall pressure) and global forces and PSD analysis.

### 4.2.1 Wake region

The wake can be separated in a near wake region, from $1 < \frac{x}{c} < 2$, and a far wake region which extents until the half of the computing domain (20 chords length). This far wake is shown in a truncated illustration in Figure 3 or Figure 4 until $\frac{x}{c} = 5$.

**Near wake region**

After the convergence is reached, the "spoiler" case shows a periodic $C_L$ and $C_D$ behaviour, whereas the "no spoiler" case
does not vary in time (see Figure 8a), as expected. For the "spoiler" case, the net area between the $C_p$ curves varies progressively in synchronicity with the $C_L$ extrema (the red dot illustrates $C_{L_{max}}$ and the blue dot corresponds to $C_{L_{min}}$ on the Figure 8b), leading to a progressive increase and decrease of the lift and drag, as illustrated in Figure 9.

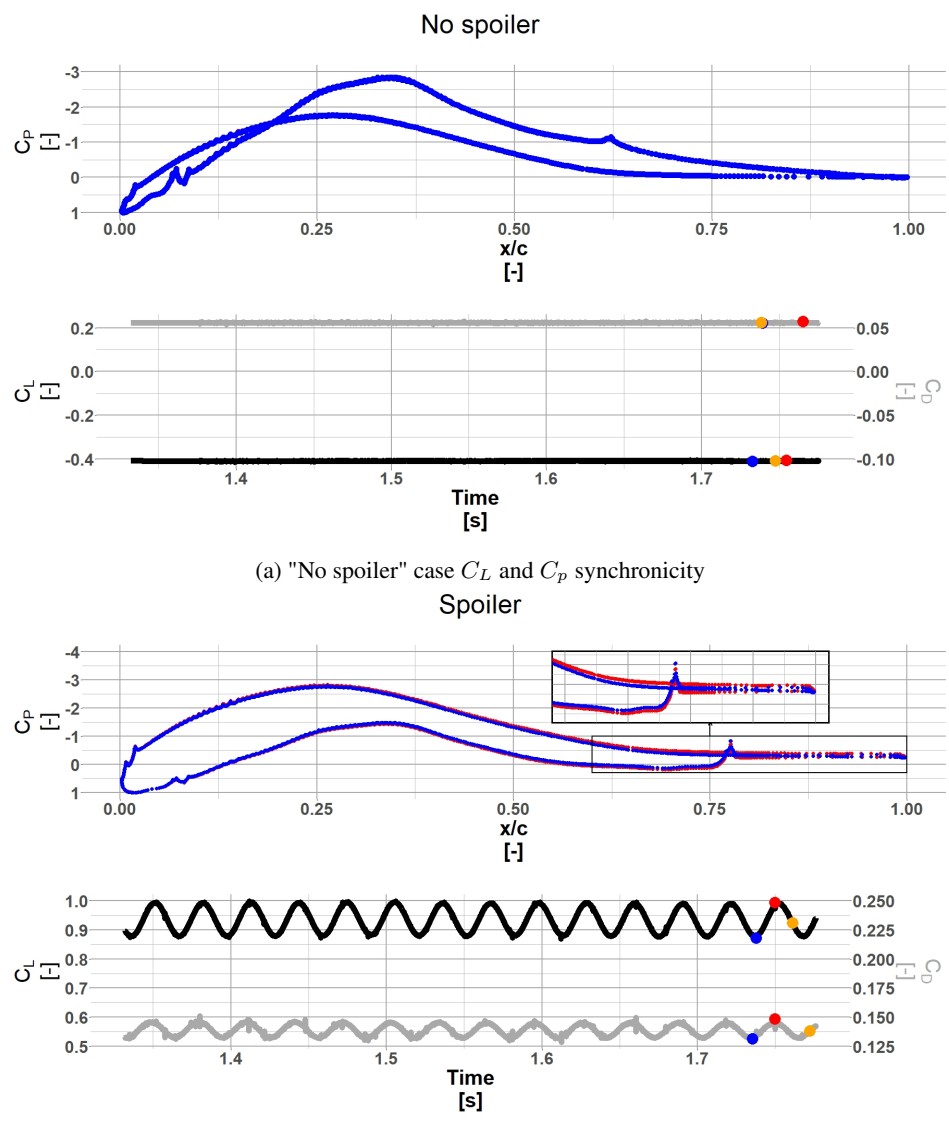

(a) "No spoiler" case $C_L$ and $C_p$ synchronicity

(b) "Spoiler" case $C_L$ and $C_p$ synchronicity

**Figure 8.** $C_p$ and aerodynamic coefficients evolution in time $\alpha = 6°$ and $Re_c = 3 \times 10^6$. For both the $C_p$ and $C_L/C_D$ plot, the blue dot (●) corresponds to the minimum $C_L$, the red dot (●) corresponds to the maximum $C_L$. On the $C_L/C_D$ plot the orange dot (●) corresponds to the mean $C_L$.

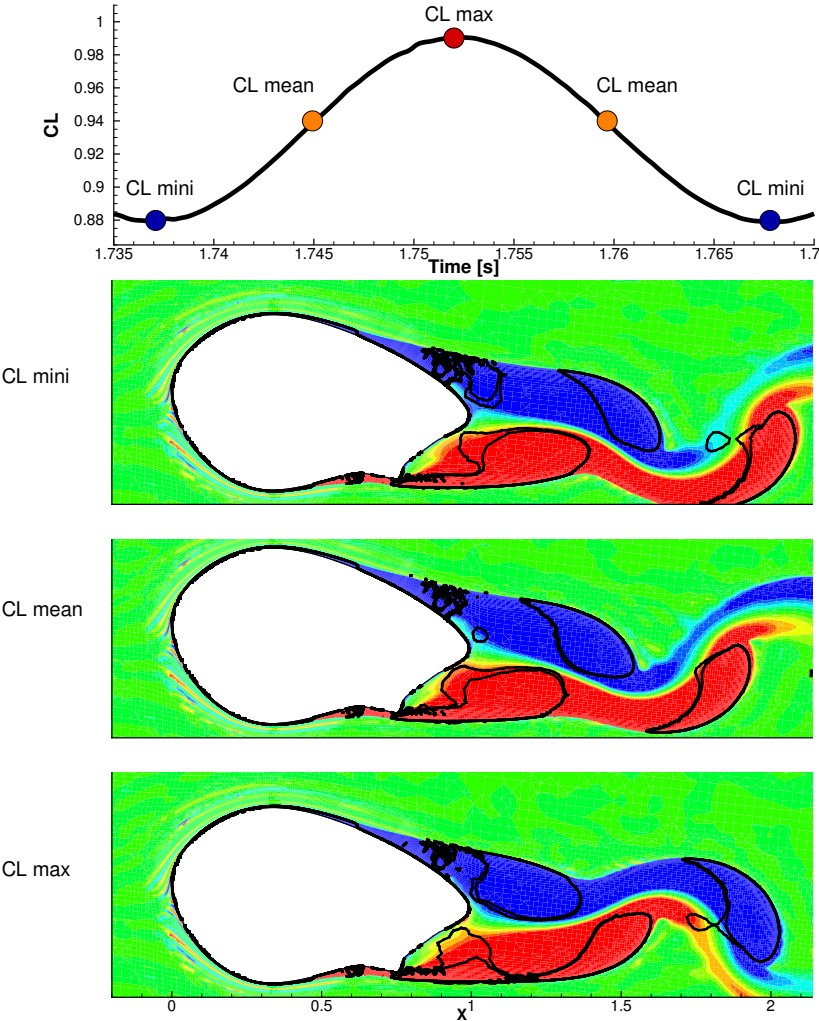

**Figure 9.** "Spoiler case" illustration of vortices in the vicinity of the trailing edge in relation with the lift coefficient evolution for $\alpha = 6°$ and $Re_c = 3 \times 10^6$. The contour plots depicts instantaneous vorticity contour with Q-criteria lines.

The near wake region is zoomed in Figure 10. It is illustrated by co-plotting a snapshot of the vorticity sign with the Q-criteria. Both instantaneous snapshots show vortices formed in the near wake region due to the roll up of the separated

shear layers from both, the upper and lower side of the aerofoil. For the "no spoiler case", the "black lines" (isolines of $0 < Q - criteria < 1000$) clearly shows that vortices are symmetric with the wake centre line. For the "spoiler case", AAO clearly starts the separation of the shear layer in the lower side of the aerofoil while the "black lines" are no longer symmetric with the wake centre line. The time evolution of this near wake organisation exhibits periodic interaction of vortices from each side of the aerofoil. The vortex from the pressure side is rolling up onto the suction side thereby forcing the upper side

separation periodically (see attached movie).

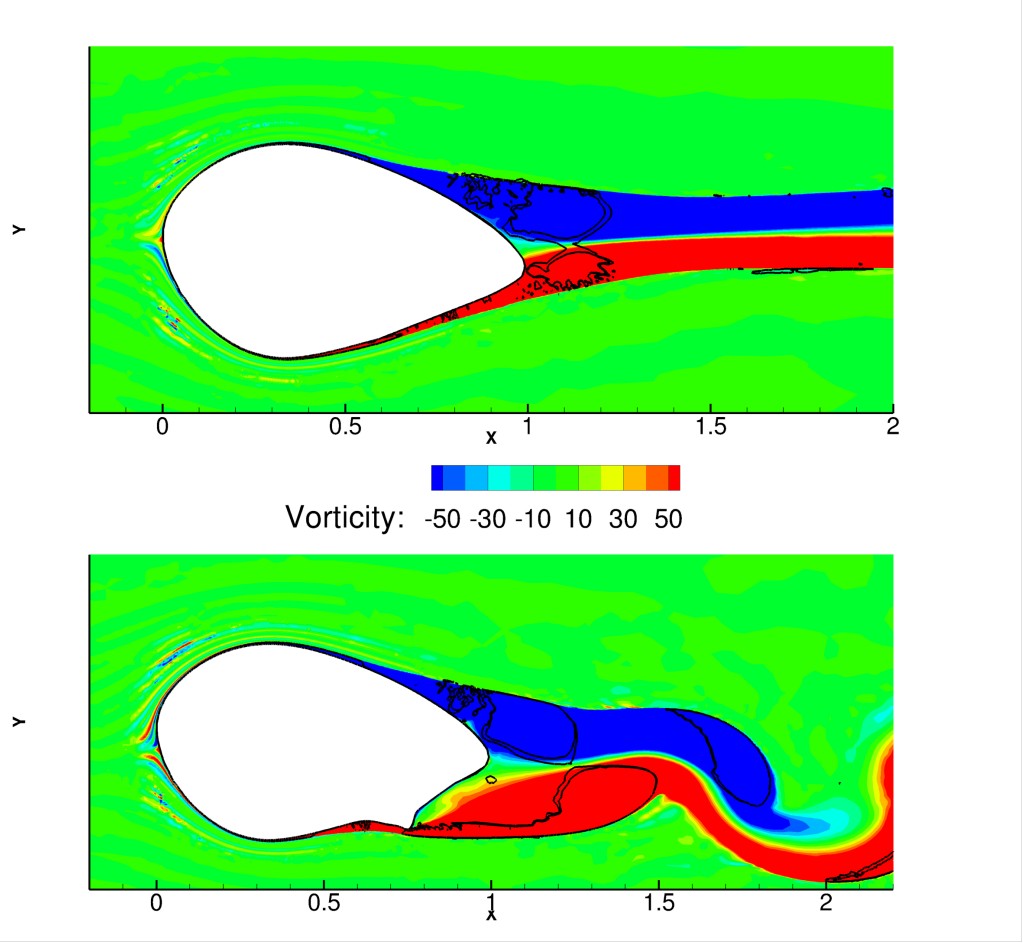

**Figure 10.** Vorticity contour plot with Q-criteria lines for $\alpha = 6°$ and $Re_c = 3 \times 10^6$, top - no spoiler case, bottom - spoiler case. Red is counterclockwise flow rotation and blue clockwise flow rotation.

In term of energy production, it is interesting to extract the snapshot of the near wake organisation at the optimal lift to drag ratio, which also occurs at $C_{L_{max}}$. For that purpose, the different $C_L$ can be analysed together with the spatial vortex organisation (see Figure 9). The lift is at its minimum (blue dots) when the top vortex is "far" from the trailing edge and the lower side vortex is rolling up toward the upper side. The mean $C_L$ (orange dots) is characterised by having both vortices close to the trailing edge: the top side vortex already separated from the surface and the lower one still attached to the spoiler's tip. Finally, the maximum lift (red dot) is seen when the lower side vortex is about to separate from the spoiler's tip and the upper side vortex is at its maximum size (it just left the aerofoil's surface). Consequently, it indicates that the pressure is at its lowest on the upper side. Therefore, in term of energy production, having shed vortices at their maximum size and the closest to the trailing edge is the best flow organisation.

**Far wake region**

In the far wake region, a single peak frequency emerges, with its harmonics, that can be extracted at $\frac{x}{c} = 3.0$ using PSD analysis (see Figure 11). The energy content for the "no spoiler" case is several order of magnitude lower than the "spoiler" case as expected.

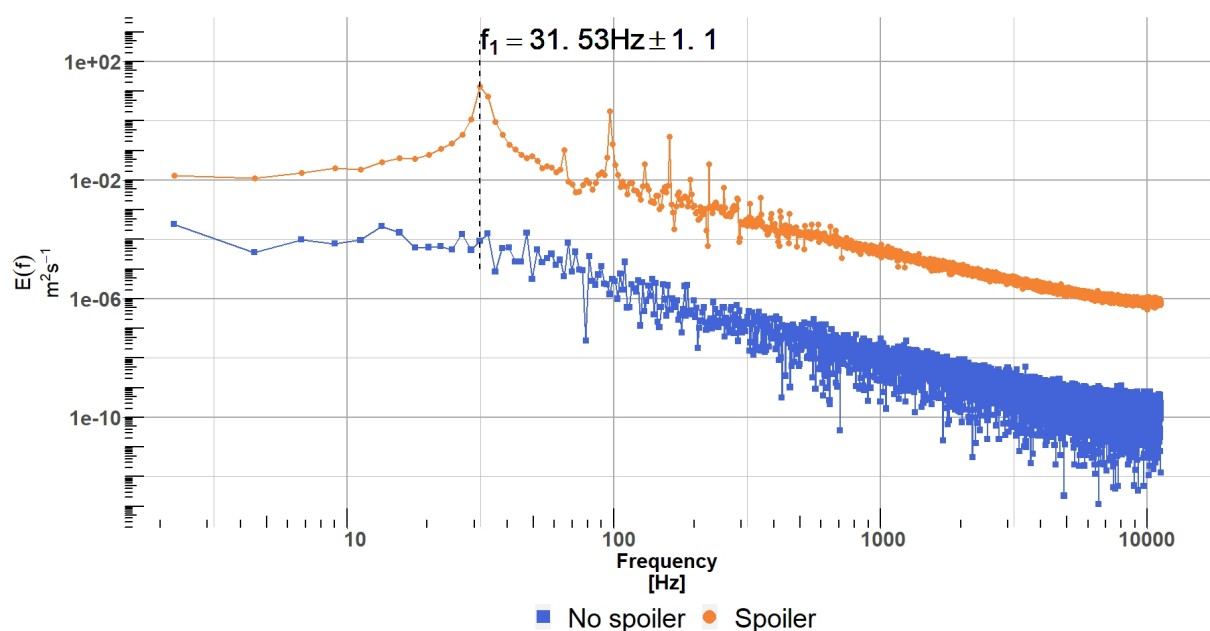

**Figure 11.** Horizontal velocity PSD, in the wake, at $\frac{x}{c} = 3.0$. The blue square (■) shows the "no spoiler" case while the orange dot (●) shows the "spoiler" case.

At last, following the definition of Yarusevych et al. (2009) a Strouhal number of $S_{t-spoiler}^* = 0.15$ is found. In this definition, the velocity used is the mean free stream velocity and the characteristic length ($L$) is the distance between two mean

horizontal velocity Root Mean Square extrema at $\frac{x}{c} = 3.0$. The RMS peaks represents the aerofoil upper side vortex centre and aerofoil lower side vortex centre, therefore the vertical distance can be viewed as a representation of the wake width. As seen in Figure 12, the "no spoiler" case does not present two distinct peaks, only a single bell-type curve representing the velocity deficit in the wake. The "spoiler" case also shows a larger velocity deficit accompanied with a pair of RMS peaks showing the presence of vortices centre.

$$St^*_{spoiler} = \frac{f \times L}{U} = \frac{31.53 \times 0.2191}{45} = 0.15 \tag{2}$$

Where $f$ is the main vortex shedding frequency, $L$ is the characteristic length and $U$ the incoming velocity.

This result falls in line with their study where $S^*_t \approx 0.17$ was found, albeit in our case at a much higher Reynolds number and for a much thicker aerofoil and equipped with spoiler.

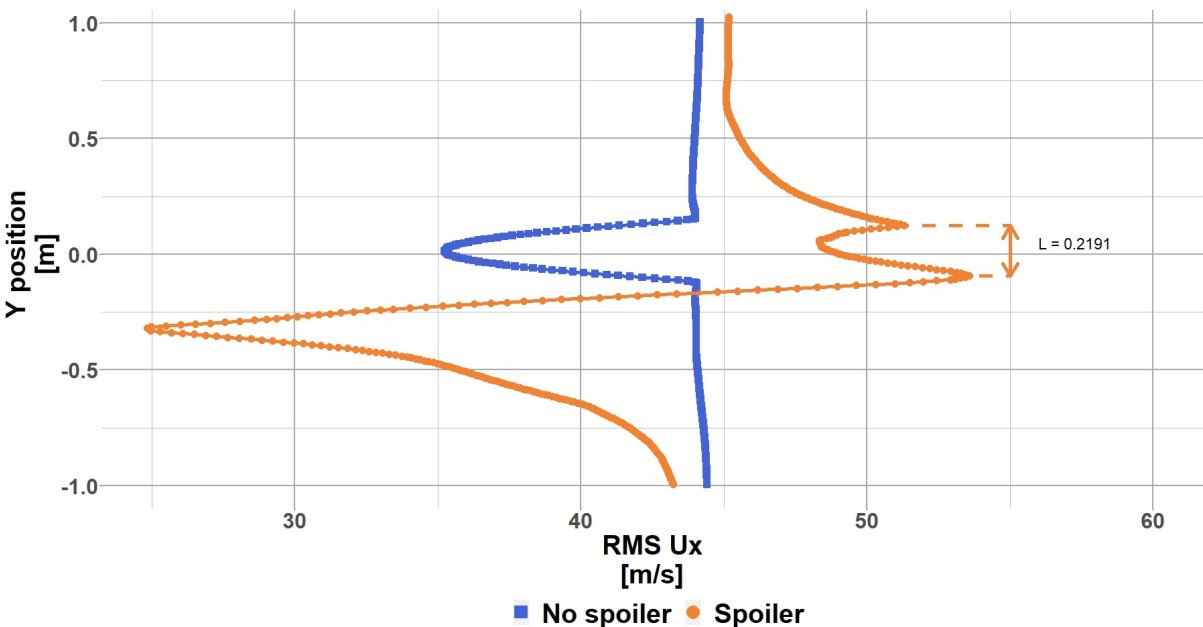

**Figure 12.** Mean Horizontal velocity Root Mean Square value for the radial position R6 at $\alpha = 6\circ$ ($\frac{r}{R} = 13\%$) and at $\frac{x}{c} = 3$. The blue square (■) shows the RMS for the no spoiler case and the orange dot (●) shows the RMS for the spoiler case.

### 4.2.2 Unsteady aerodynamic polar

This unsteady analysis for $\alpha = 6°$ case is extended towards all other angles of attack. The vortex shedding organisation previously described induces oscillations on the surface pressure and thereby the aerodynamic coefficients $C_L$ and $C_D$.

The behaviour described for the angle of attack $\alpha = 6°$ is present throughout the polar for both cases with varying amplitude of the oscillations. The same analysis was carried out for $\alpha = 10°$ (not presented in this paper), the vortex shedding frequency remains similar, only the amplitude changes. Overall, as long as the angle of attack is in the linear region the vortex shedding frequency remains similar, when approaching stall or in the stall region the frequency drops by half the attached flow region value. The maximum and minimum of these oscillations are reported in Figure 13 and Figure 14. The "no spoiler" case shows a decreasing variation of lift and drag coefficients from $-20° < \alpha < -2°$. The variation remains constant until the higher angles of attack ($\alpha > 10°$). The variation in aerodynamic coefficients for the "spoiler" case is similar for the negative angle of attack and in the linear region. It increases drastically after $10°$ showing a possible stall behaviour, as highlighted by the coloured areas. Beyond $14°$ for the $C_L$, the variation amplitude is similar to the actual mean aerodynamic value. Overall, the "spoiler" case adds a lot more variation in the aerodynamic loads when it becomes efficient (i.e. the lift is increased).

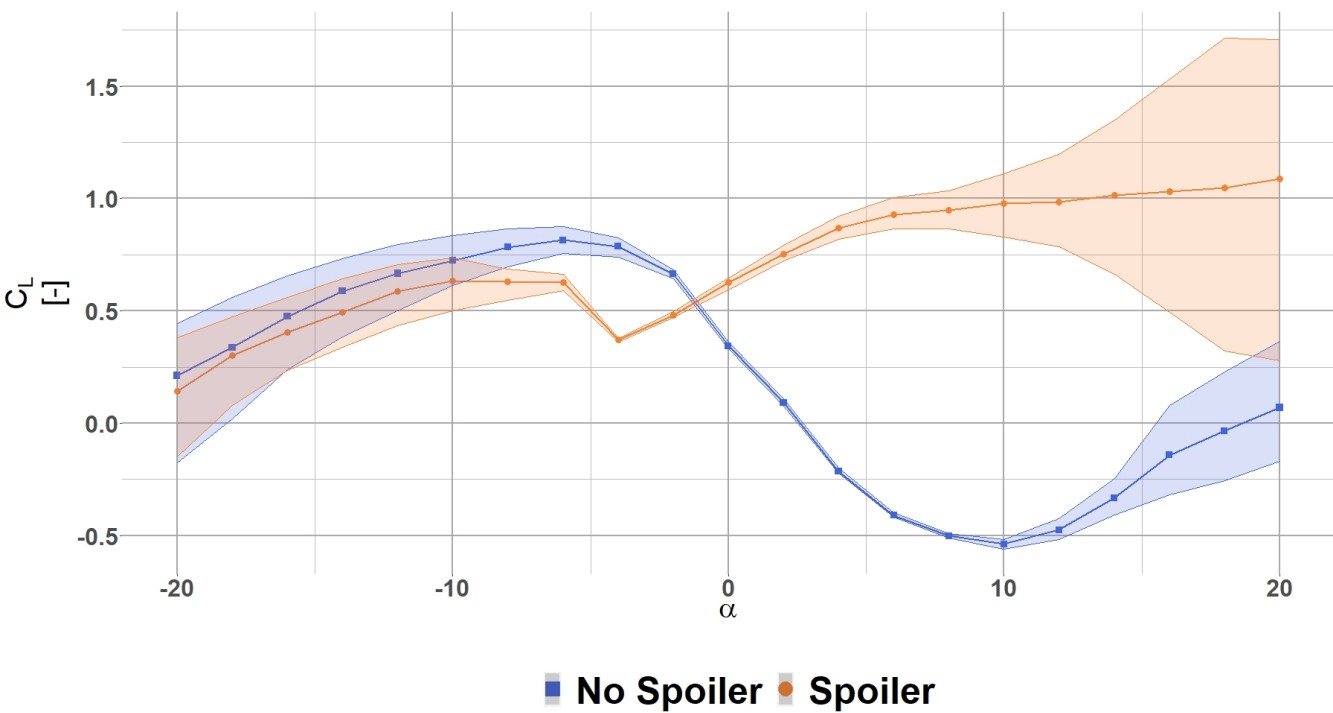

**Figure 13.** Lift coefficient polar for the radial position R6 ($\frac{r}{R} = 13\%$). The blue square (■) shows the $C_L$ for the no spoiler case and the orange dot (●) shows the $C_L$ for the spoiler case.

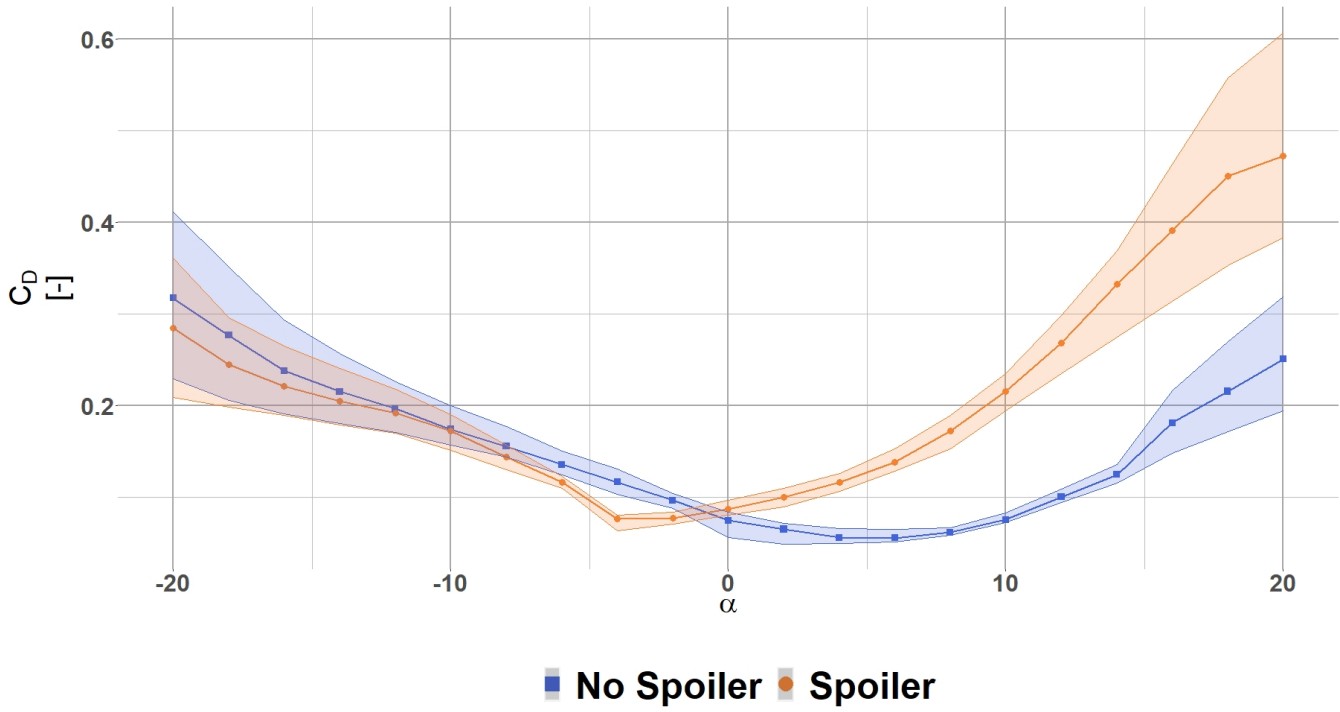

**Figure 14.** Drag coefficient polar for the radial position R6 ($\frac{r}{R} = 13\%$). The blue square (■) shows the $C_D$ for the no spoiler case and the orange dot (●) shows the $C_D$ for the spoiler case.

## 5  Conclusions

The present study proposes an original 2D URANS description of the unsteady flow behind thick aerofoil (59%) from an operating 2MW wind turbine equipped with spoilers at high Reynolds number. The particularity of this configuration stands on the size of the AAO: it is the real dimension of today's operating wind turbine rather than the sub-boundary layer device sizes usually studied in the literature. This AAO is found to efficiently rearrange the mean flow, adding lift throughout the positive angles of attack. However, the drawback is a high drag penalty coupled with high unsteadiness of the aerodynamic forces. Without spoiler, the aerofoil wake is erratic and not organised. With spoiler, a peak frequency is dominant in the aerodynamic lift and drag coefficients, which corresponds to a vortex shedding organisation. The associated Strouhal number is almost constant $St^* = 0.17$ despite the aerodynamic coefficients variation amplitude changing with the angle of attack. The wake energy content is increased by several order of magnitude compared to the aerofoil without spoiler. This flow reorganisation is present throughout the polar range and is accompanied by larger variation of aerodynamic forces than without spoiler. The impact of this type of excitation will be quantified further in term of energy production and fatigue in future work.

*Code and data availability.*  Available on demand.

*Video supplement.*  A short movie depicting the vortex shedding for the "spoiler" case is available Potentier (2021).

*Author contributions.*  TP performed the scans post-processing, CFD pre-processing and post-processing, and writing of the paper. EG performed CFD verification and helped set-up the CFD model. ClB and AF provided feedback from the industrial point of view and CB helped with the proofreading of the manuscript and physical analysis of the results.

*Competing interests.*  The authors declare that they have no conflict of interest.

*Acknowledgements.*  The authors would like to acknowledge the ANRT (Association Nationale de Recherche Technologique) for their financial support.

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
