# Peer review of "High Reynolds number wind turbine blade equipped with root spoilers. Part I: Unsteady aerodynamic analysis using URANS simulations."

_Wind Energy Science, 2021_

## Author Response (AR1)

**High Reynolds number wind turbine blade equipped with root spoilers. Part I: Unsteady aerodynamic analysis using URANS simulations**

**Reviewer 1**

The Authors performed an interesting study on the aerodynamic performance of a very thick airfoil equipped with a passive device to be adopted on the root region of a wind turbine. The results show the dramatic change in the airfoil performance in comparison with the smooth airfoil. The results are well presented, the paper is well written and the methodology is suitable for the purpose of the work. The novelty of the work relies on the high thickness of the studied airfoil, which is not commonly found in literature.

The Reviewer has the following comments:

- The abstract is very short and does not give a clear overview on the contents of the paper. In particular, it is not clear that the simulation is focused only on the 2D simulation of a thick airfoil of the root region of the blade. It is simply stated "A wind turbine blade equipped with root spoilers is analysed using 2D…" that is a too general sentence.
- The authors did not show a validation of the numerical model against experimental data. Since the thickness of the airfoil is definitely higher than the standard (it is almost a bluff body), the authors should provide some information regarding the accuracy of the proposed method.
- Which is the definition of the Strouhal adopted in the present case? Is it based on the blade chord or on the spoiler thickness?
- Which is the temporal resolution adopted for the unsteady calculations?

**Response to Reviewer 1**

We thank the reviewer for the time spent reviewing the article and the feedback given. Below are the answer to the questions asked.

1. The abstract has been re-written to be longer and make it clear that the paper is about 2D CFD simulations and analysis:

A commercial wind turbine blade equipped with root spoilers is analysed using 2D URANS Computational Fluid Dynamics (CFD) to assess the unsteady impact of passive devices. In this work, we present the 2D CFD unsteady results from a non-rotating single thick section located at the root end of the blade with and without spoiler. Computations were performed at the chord-based Reynolds number $Re\_c = 3 \times 10^6$. The analysed spoiler is of commercial size with a height of approximately 33\% of the local chord. Comparatively to the existing literature, it is at least one order of magnitude larger than the size of the well known Gurney flaps. The analysis is first performed in the steady state at a single angle of attack using global aerodynamic forces, the local pressure distributions and flow field analysis. It shows a very important flow rearrangement in the presence of a spoiler, responsible of the mean lift force enhancement. Then, simulations are extended over a large range of angle of attack (from $-20^\circ$ to $20^\circ$), to identify the spoiler behaviour across the polar. Analyses are then continued accounting for the flow unsteadiness. The spoiler induces an important wake behaviour linked to the apparition of global load fluctuations. Using the wall pressure distributions and the associated spatio-temporal organisation of the flow field those fluctuations are well characterised. The detailed analysis performed at one angle of attack, is then extended to assesses the load fluctuations at other angles of attack, showing the evolution of the unsteady loads in relation to the wake

2. Unfortunately such experimental data does not exist to the best of our knowledge, therefore we could not validate our simulations against wind tunnel measurements. However, in Potentier et al. [1] a validation of the code has been done against the existing DANAERO database for aerofoil as thick as 36%, giving confidence in our approach and results.

[1]: T. Potentier, E. Guilmineau, and C. Braud, 'CFD solver and meshing techniques verification using the DANAERO database', p. 8, 2020.

3. The Strouhal number is calculated using the definition of Yarusevych et al. [2]. As detailed on page 16 line 178 of our manuscript, the characteristic length (L) is the distance between two peaks of RMS of the horizontal velocity. The paragraph has been modified as follow :

In the far wake region, a single peak frequency emerges, with its harmonics, that can be extracted at $\frac{x}{c} = 3.0$ using PSD analysis (see Figure \ref{PSD_Ux_R6}). The energy content for the "no spoiler" case is several order of magnitude lower than the "spoiler" case as expected.

[Figure]

Figure 11. Horizontal velocity PSD, in the wake, at $\frac{x}{c} = 3.0$. The blue square (\textcolor{blue}{\large$ \blacksquare $}) shows the "no spoiler" case while the orange dot (\textcolor{orange}{\Large$ \bullet $}) shows the "spoiler" case.

At last, following the definition of \cite{yarusevychVortexSheddingAirfoil2009} a Strouhal number of $S_{t-spoiler}^* = 0.15$ is found. In this definition, the velocity used is the mean free stream velocity and the characteristic length ($L$) is the distance between two mean horizontal velocity Root Mean Square extrema at $\frac{x}{c} = 3.0$. The RMS peaks represents the aerofoil upper side vortex centre and aerofoil lower side vortex centre, therefore the vertical distance can be viewed as a representation of the wake width. As seen in Figure \ref{PMS_Ux_R6}, the "no spoiler" case does not present two distinct peaks, only a single bell-type curve representing the velocity deficit in the wake. On the other hand, the "spoiler" case shows a larger velocity deficit accompanied with a pair of RMS peaks showing the presence of vortices centre.

\begin{equation}

St^*_{spoiler} = \frac{f \times L}{U} = \frac{31.53 \times 0.2197}{45} = 0.15

\end{equation}

Where $f$ is the main vortex shedding frequency, $L$ is the characteristic length and $U$ the incoming velocity.\\

This result falls in line with their study where $S_t^* \thickapprox 0.17$ was found, albeit in our case at a much higher Reynolds number and for a much thicker aerofoil and equipped with spoiler.

[Figure]

Figure 12. Mean Horizontal velocity Root Mean Square value for the radial position R6 at $\alpha = 6\circ$ ($\frac{r}{R}=13\%$) and at $\frac{x}{c} = 3$. The blue square ($\textcolor{blue}{\large \blacksquare}$) shows the RMS for the no spoiler case and the orange dot ($\textcolor{orange}{\Large \bullet}$) shows the RMS for the spoiler case.

[2]: S. Yarusevych, P. E. Sullivan, and J. G. Kawall, 'On vortex shedding from an airfoil in low-Reynolds-number flows', *J. Fluid Mech.*, vol. 632, pp. 245–271, Aug. 2009, doi: 10.1017/S0022112009007058.

4.  After a time step convergence study based on the chosen mesh (not initially presented in the manuscript for conciseness) we chose to use a time step of $\frac{1}{500}$ of the time necessary for an air particle to travel the chord distance. For the aerofoil and velocity used (45m/s) in the present manuscript, it is equivalent to $\Delta t = 4.44 \times 10^{-5} s$. The difference between the chosen time step and the finest one is small enough so that the calculation time benefit outweighs the small uncertainty introduced. The following will be added to the revised manuscript:

A time step convergence study using the "Fine grid" has also been performed and summarised in Table 3.

The chosen time step for the rest of the study is Δt=4.44 ×10^(-5) s because of the good balance between result accuracy and time prior convergence.

Table 3. Time step independence study for the scanned blade with spoiler at α = 0° and Re$_c$ = 3 ×10$^6$

| Time step [s] | CL [-] | Extrapolated CL error [-] | CD [-] | Extrapolated CD error [-] | Time before convergence [min] |
|---|---|---|---|---|---|
| 2.22E-04 | 0.596 | -5.70% | 0.08280 | -6.18% | 2719 |
| 8.89E-05 | 0.599 | -5.17% | 0.08289 | -6.08% | 3028 |
| 4.44E-05 | 0.619 | -1.99% | 0.08705 | -1.36% | 3709 |
| 2.22E-05 | 0.628 | -0.54% | 0.08809 | -0.17% | 11118 |
| Richardson extrapolation | 0.632 | 0.00% | 0.08825 | 0.00% | N/A |

**Reviewer 2**

The paper presents an investigation on the effects of using spoilers at the root of a turbine blade to improve the aerodynamic performance of the profile. The study is interesting and introduces new information, the topic is relevant for the scopes of Wind Energy Science.

The analysis is limited to 2D results, both steady and time-resolved; the problem is properly introduced and the results are nicely presented and discussed. The referee has found some lacking aspects in the description of the methodology. For this reason, the referee is recommending a revision of the paper prior to consider the paper suitable for publication.

1) In section 3.1, where the CFD code is presented, the authors provide the details of the discretization scheme only for the unsteady term, while full details would be requited about source, divergence and Laplacian terms appearing in the equations (included the ones related to turbulence). The solution strategy should also be reported (SIMPLE, PISO, etc.).

2) In section 3.2 the verification of the CFD code is reported, with convincing arguments. However, this computational study completely misses a proper validation of the computational tool, considering both steady and unsteady results as well as near and far wake predictions. Since this is a journal publication, the authors are encouraged to provide the experimental assessment of the code or, at least, introduce relevant references and a brief discussion.

3) The results are presented in detail with the support of references, and identifying the novelty in the approach. There is only one aspect which is questionable, related to the far wake analysis; as well known, U-RANS approaches might face difficulties in reproducing reliably the far wake of blades (and wind turbines); LES approaches might be preferable, especially considering that the wake of the blade equipped with spoilers is affected by aerodynamic instabilities. In absence of a proper validation, how reliable can be considered the spectra reported in Figure 11? Authors should discuss this issue in the paper.

**Response to Reviewer 2**

We thank the reviewer for the quality of the feedback. Below are the answers to the several points noted.

1.  It is noted that ISIS-CFD code has not been properly cited in the current state of the paper. Therefore we have updated the section 3.1 :

ISIS-CFD, developed by Centrale Nantes and CNRS and available as a part of the FINE\texttrademark/Marine computing suite, is used in the present study to solve the incompressible Unsteady Reynolds-Averaged Navier-Stokes (URANS) equations. It is based on the finite volume method to build the spatial discretization of the transport equations. The unstructured discretization is face-based, which means that cells with an arbitrary number of arbitrarily shaped faces are accepted (unstructured mesh). A second order backward difference scheme is used to discretize time. The solver can simulate both steady and unsteady flows. In the case of turbulent flows, transport equations for the variables in the turbulence model are added to the discretization. \\

All flow variables are stored at the geometric cent of arbitrary shaped cells. Volume and surface integrals are evaluated with second-order accurate approximations. The method is face-based, which means that the net fluxes in the cells are computed face by face. Thus, the cells with an arbitrary number of arbitrarily shaped faces are accepted. Numerical fluxes are reconstructed on the mesh faces by linear extrapolation of the integrand from the neighbouring cell centres. A centred scheme is used for the diffusion terms, whereas for the convective fluxes, a blended scheme with 80\% central and 20\% upwind is used.\\

The velocity field is obtained from the momentum conservation equations and the pressure field is extracted from the mass equation constraint, or continuity equation, transformed into a pressure equation. The pressure equation is obtained by the Rhie and Chow interpolation \cite{rhie_chow_83}. The momentum and pressure equations are solved in an segregated manner as in the SIMPLE coupling procedure \cite{issa_85}. A detailed description of the discretisation is given by \cite{queutey_visonneau_07}.\\

The turbulence model used is SST k-$\omega$ (see \cite{menterZonalTwoEquation1993}). The flow characteristics are representing the air at sea level at a temperature of \unit{15}{°C}, i.e.: $\nu$ = 1.81$\times10^{-5}$\unit{}{kg m$^{-1}$ s$^{-1}$} (dynamic viscosity) and $\rho$ = \unit{1.225}{kg m$^{-3}$} (air density). A uniform inflow of \unit{45}{m/s} is set, which induces a chord Reynolds number of: $Re_c = 3\times 10^6$ for the presented section of $\frac{r}{R}=13\%$.

1.  ISIS-CFD has been validated many times over regarding 3D cases for hydrodynamics and automotive cases (see [4, 5, 6, 7, 8]). However, the 2D cases validation were not explicitly cited or showed in the present paper. As part of [10], a comparison between the existing DANAERO literature both experimental and numerical and ISIS-CFD was performed. The following figures illustrates the good agreement between the 2D wind tunnel experiment and the 2D URANS ISIS CFD simulations.

[Figure]

DANAERO Section03 and Section05

[Figure]

DANAERO Section08 and Section10

The following lines will be added to the introduction of the section 3.2:

"A comparison between the existing DANAERO literature both experimental and numerical and ISIS-CFD was performed (see \cite{potentier_cfd_2020}). The outcome showed the good agreement between the 2D wind tunnel experiment and the 2D URANS ISIS CFD simulations, thus validating the use of ISIS-CFD for 2D external applications. The domain size study has also been performed and the recommended square domain of 80 chords in length was used."

[1] C.M. Rhie, and W.L. Chow, 'A numerical study of the turbulent flow past an isolated aerofoil with trailing edge separation', AIAA Journal, vol. 17(11)  pp. 1525–1532, 1983, doi: 10.2514/6.1982-998.

[2] R. Issa, 'Solution of the implicitly discretized fluid flow equations by operator-splitting', Journal of Computational Physics, vol. 62, pp 40–65, 1985, doi: 10.1016/0021-9991(86)90099-9.

[3] P. Queutey and M. Visonneau, 'An interface capturing method for free-surface hydrodynamic flows', Computers and Fluids, vol. 36, pp. 1481–1510, 2007,  doi: 10.1016/j.compfluid.2006.11.007.

[4] J. Wackers *et al.*, 'Can adaptive grid refinement produce grid-independent solutions for incompressible flows?', *Journal of Computational Physics*, vol. 344, pp. 364–380, Sep. 2017, doi: 10.1016/j.jcp.2017.04.077.

[5] J. Wackers, G. Deng, E. Guilmineau, A. Leroyer, P. Queutey, and M. Visonneau, 'Combined refinement criteria for anisotropic grid refinement in free-surface flow simulation', *Computers & Fluids*, vol. 92, pp. 209–222, Mar. 2014, doi: 10.1016/j.compfluid.2013.12.019.

[6] E. Guilmineau, G. B. Deng, A. Leroyer, P. Queutey, M. Visonneau, and J. Wackers, 'Numerical Simulations for the Wake Prediction of a Marine Propeller in Straight-Ahead Flow and Oblique Flow', *Journal of Fluids Engineering*, vol. 140, no. 2, p. 021111, Feb. 2018, doi: 10.1115/1.4037984.

[7] G. B. Deng, E. Guilmineau, P. Queutey, and M. Visonneau, 'Ship Flow Simulations with the ISIS CFD Code', *CFD Workshop Tokyo 2005 (CFDWS2005)*, p. pp 474--482, Mar. 2005.

[8] E. Guilmineau, G. B. Deng, A. Leroyer, P. Queutey, M. Visonneau, and J. Wackers, 'Assessment of hybrid RANS-LES formulations for flow simulation around the Ahmed body', Computers & Fluids, vol. 176, pp. 302–319, Nov. 2018, doi: 10.1016/j.compfluid.2017.01.005.

[9] M. Visonneau, E. Guilmineau and S. Toxopeus, 'Study on the onset and development of the three-dimensional separation around a diamond wing with incompressible flow solvers', Aerospace Science and Technology, vol. 57, pp. 52-75, 2016, doi: 10.1016/j.ast.2016.02.012.

[10] T. Potentier, E. Guilmineau, and C. Braud, 'CFD solver and meshing techniques verification using *the DANAERO database', p. 8, 2020.*

2.  This is indeed an interesting question that can be answered only by comparing 2D URANS simulations and LES simulations. However, LES simulations of this present configuration do not exist in the literature and is therefore a full work by itself. We believe that the present URANS simulations have a sufficient representation of the wake dynamics (until x/c = 5.0) to perform a first analysis of the spoiler impact. Ongoing RANS-DES simulations of the present configuration will be used in the future as a comparison with the present URANS simulations, which is out of the scope of the present paper.